# Comparative Genomic Analysis Uncovers the Chloroplast Genome Variation and Phylogenetic Relationships of *Camellia* Species

**DOI:** 10.3390/biom12101474

**Published:** 2022-10-13

**Authors:** Ping Lin, Hengfu Yin, Kailiang Wang, Haidong Gao, Lei Liu, Xiaohua Yao

**Affiliations:** 1State Key Laboratory of Tree Genetics and Breeding, Research Institute of Subtropical Forestry, Chinese Academy of Forestry, Hangzhou 311400, China; 2Key Laboratory of Tree Breeding of Zhejiang Province, Research Institute of Subtropical Forestry, Chinese Academy of Forestry, Hangzhou 311400, China; 3Genepioneer Biotechnologies Co., Ltd., Nanjing 210023, China

**Keywords:** *Camellia*, chloroplast genome, genetic relationship, marker, phylogenetic analysis

## Abstract

*Camellia* is the largest genus in the family Theaceae. Due to phenotypic diversity, frequent hybridization, and polyploidization, an understanding of the phylogenetic relationships between *Camellia* species remains challenging. Comparative chloroplast (cp) genomics provides an informative resource for phylogenetic analyses of *Camellia*. In this study, 12 chloroplast genome sequences from nine *Camellia* species were determined using Illumina sequencing technology via *de novo* assembly. The cp genome sizes ranged from 156,545 to 157,021 bp and were organized into quadripartite regions with the typical angiosperm cp genomes. Each genome harbored 87 protein-coding, 37 transfer RNA, and 8 ribosomal RNA genes in the same order and orientation. Differences in long and short sequence repeats, SNPs, and InDels were detected across the 12 cp genomes. Combining with the complete cp sequences of seven other species in the genus *Camellia*, a total of nine intergenic sequence divergent hotspots and 14 protein-coding genes with high sequence polymorphism were identified. These hotspots, especially the InDel (~400 bp) located in *atpH*-*atpI* region, had sufficient potential to be used as barcode markers for further phylogenetic analysis and species identification. Principal component and phylogenetic analysis suggested that regional constraints, rather than functional constraints, strongly affected the sequence evolution of the cp genomes in this study. These cp genomes could facilitate the development of new molecular markers, accurate species identification, and investigations of the phylogenomic relationships of the genus *Camellia*.

## 1. Introduction

*Camellia*, comprising more than 200 species [1], is the largest genus in the family Theaceae, with remarkable economical and phylogenetic value [2]. *Camellia* plants offer abundant species diversity and include several economically important members. For instance, species in the section (sect.) *Oleifera*, including *C. oleifera* and *C. meiocarpa*, have a long history of cultivation and utilization in China for the acquisition of high-quality edible oil. Species belonging to the sect. *Thea*, including *C. sinensis* and *C. assamica*, provide the world’s oldest and most popular caffeine-containing tea beverage [3]. Moreover, species in the sect. *Camellia* have great ornamental value, particularly *C. japonica*. Some other *Camellia* species, such as *C. chekiangoleosa*, *C. semiserrata*, etc., have the additional benefit of bearing beautiful flowers. Because of such a variety of uses, *Camellia* plants are grown all over the world [2].

Due to phenotypic diversity, frequent hybridization, and polyploidization, a full understanding of the phylogenetic relationships of the genus *Camellia* is quite problematic [4,5,6]. Phenotypes are often affected by environmental factors, complicating morphology-based traditional phylogenetic analysis. Nuclear genome sequencing has frequently been used to analyze phylogenetic relationships, genetic diversity, and evolutionary pathways in the genus *Camellia* to gain insights into the phylogeny of these species [7,8,9,10].

The angiosperm chloroplast (cp) genomes that are maternally transmitted with relatively stable structure [6,11] could provide an informative and valuable resource for phylogenetic analyses at family/genus/species levels [12,13,14]; they are also suitable for the resolution of complex evolutionary relationships [15,16,17] and trace source populations [18,19,20]. In the past decades, cp genomes have proven to be particularly powerful in revealing phylogenetic relationships of *Camellia* plants, which have large nuclear genomes [3,21,22]. Cp-derived markers were successfully employed to study the evolutionary relationships of plants in sects. *Thea*, *Camellia*, *Theopsis*, *Stereocarpus*, *Chrysantha*, and *Corallina*, etc., providing valuable genetic information for accurate identification of species and reconstruction of the phylogeny of the genus *Camellia* [6,11,12,13,14,23,24,25,26,27]. However, the lack of suitable polymorphic genetic markers has obstructed the phylogenic analysis of sect. *Oleifera*, leaving its taxonomic classification rather controversial. For example, *C. vietnamensis*, a species defined in Zhang et al.’s taxonomic classification [5], was classified into a taxonomic variety of *C. oleifera* in Min et al. classification [1]. In turn, a better understanding and utilization of the diversification and evolution of the sect. *Oleifera* is missing. Recent advances in next-generation sequencing techniques and bioinformatics tools have made it convenient to obtain cp genome sequences at relatively low costs.

In this study, we present the complete cp genome sequences of 12 individuals from nine species of *Camellia* using the Illumina sequencing platform. The sequenced individuals covered six species from the sect. *Oleifera*, two species from the sect. *Camellia*, and one species from the sect. *Paracamellia*. Combining with the complete cp sequences of seven other species (Appendix A) in the genus *Camellia*, we performed a comparative analysis of cp genomes and identified highly variable regions across species. Combining with 20 reported complete cp genomes in the family Theaceae (Appendix A), the new phylogenetic trees were constructed. This study aims to examine variable regions in the *Camellia* cp genomes, especially in the sect. *Oleifera*, to establish a molecular basis for the development of novel DNA markers and reconstruction of phylogenetic relationships among the representative species. Our results provide a robust genomic framework for the phylogenomic characterization of *Camellia* species. It also gives a foundation to develop DNA markers that allow the identification of individual taxa cost-effectively.

## 2. Materials and Methods

### 2.1. Plant Materials

Tender leaves of Camellia plants, including *C. oleifera* var. *40*, *53*, *GR3*, *MY6*, *C. meiocarpa*, *C. nanyongensis*, *C. sasanqua*, *C. vietnamensis*, *C. gauchowensis*, *C. chekiangoleosa* var. *Baihua*, *C. semiserrata*, and *C. grijsii*, used in this study were harvested from Research Institute of Subtropical Forestry, Chinese Academy of Forestry (Hangzhou, Zhejiang, China), International Camellia Species Garden (Jinhua, Zhejiang, China), and Dongfanghong Forest Farm of Zhejiang Province (Jinhua, Zhejiang, China) in April 2020 (Table 1). The collected plant materials were classified by Sealy’s taxonomic treatment [5] (Table 1). All the samples were collected according to local, national, or international guidelines and legislation. We complied with the IUCN Policy Statement on Research Involving Species at Risk of Extinction and the Convention on the Trade in Endangered Species of Wild Fauna and Flora during the sample collection.

### 2.2. DNA Sequencing and Chloroplast Genome Assembly

About 20 g of tender leaves from each taxon were used for total genomic DNA extraction using the TaKaRa MiniBEST Plant Genomic DNA Extraction Kit (TaKaRa, Dalian, China) according to the user manual. High-quality genomic DNA was used to construct 350 bp Illumina HiSeq libraries following the manufacturer’s protocol. A paired-end 150 bp sequencing strategy was performed for each library on the Illumina NovaSeq6000 platform. Raw data were filtered using FASTp v.0.20.0 [28] to obtain clean data by trimming adapter sequences and removing low-quality reads with a Phred quality threshold *Q* < 20. We assembled the cp genome in the following steps: First, the clean reads were assembled into contigs using SPAdes v3.11.1 [29] with a range of iterative *k*-mers sizes of 55, 87, and 121. If SPAdes was unsuccessful at assembling complete cp genomes, we used the following steps to assemble it. Second, the contigs were aligned to the reference cp genome of *C. crapnelliana* (NCBI accession number NC_024541.1) using BLAST [30], and the aligned contigs were extracted, ordered, and the reads of nuclear origins were excluded. The extracted contigs were further assembled to scaffolds using SSPACE v2.0 [31]. Third, clean data were again mapped to the assembled draft cp genomes to verify the assembly results. The majority of gaps were filled by Gapfiller v2.1.1 [32,33].

### 2.3. Genome Annotation and Visualization

Assembled cp genomes from all taxa were annotated. Prodigal v2.6.3 [34] and Hmmer v3.1b2 [35] were applied to annotate protein-coding genes and rRNAs, respectively. tRNAs were predicted via ARAGORN v1.2.38 [36]. The annotation results were manually corrected for the codon positions and intron/exon boundaries by comparing to the homologous genes from *Camellia* with known cp genomes. The circular maps of the cp genome were drawn using the OGDRAW tool [37]. The exact boundaries of IR/ LSC and IR/ SSC were confirmed by aligning them with the homologous sequences from other *Camellia* species and visualized by a customized Perl script (https://github.com/xul962464/perl-IRscope (accessed on 12 June 2021)), which was similar to IRscope.

### 2.4. Sequence Divergence Analysis

Forward, reverse, palindromic, and complement LSRs were found and analyzed using vmatch v2.3.0 [38], with the parameters’ minimum length = 30 bp and Hamming distance = 3 in the cp genomes. Reverse and complement repeats were further checked by a customized Perl script. SSRs were detected by MISA v1.0 [39] with the parameters set at ≥8 repeats for mononucleotides, ≥5 repeats for dinucleotides, and ≥3 repeats for trinucleotides, tetranucleotides, pentanucleotides, and hexa-nucleotides. Whole-genome alignments were conducted to evaluate rearrangements and substantial sequence divergence using Mauve [40] and MAFFT v7.0 [41] with default parameters. Based on the whole-genome alignments by MAFFT v7.0, the SNPs and InDels calling was performed using DnaSP 5.0 [42]. Furthermore, to identify the divergent hotspots, the nucleotide diversity (*Pi*) values for each gene in the cp genomes were evaluated using DnaSP 5.0.

### 2.5. Selection Pressure Analysis

To evaluate the role of selection on the protein-coding gene regions in the cp genomes of *Camellia* species, we calculated the nonsynonymous mutation rate (*Ka*), synonymous mutation rate (*Ks*), and the *Ka*/*Ks* of each gene from the 12 *Camellia* taxa in this study and the seven related *Camellia* species downloaded from NCBI GenBank (Appendix A) using the *KaKs*_Calculator v2.0 [43].

### 2.6. Principal Component Analysis (PCA) and Phylogenetic Analysis

SNP data of the 12 cp genomes sequenced in this study were used to perform PCA using GCTA v1.25.2 [44], and the first two components were plotted. The complete cp genome sequences of 26 *Camellia* taxa were used for phylogenetic analyses, including 12 cp genomes reported in this study and 14 previously sequenced cp genomes downloaded from the NCBI GenBank. Another six cp genomes from the genera *Tutcheria*, *Apterosperma*, *Stewartia*, *Anneslea*, *Adinandra*, and *Pyrenaria* in Theaceae were selected as outgroup. The whole-genome alignments of 32 species or their variations were performed using the MAFFT v7.0 program [41]. The aligned results were trimmed by trimAI v1.4. The phylogenetic analyses were implemented using maximum likelihood (ML) and Bayesian analysis methods based on the complete cp genome data, coding region data and noncoding region data, respectively. ML analyses were implemented in RAxML v.8.2.10 [45] and Bayesian analysis was conducted in MrBayes v.3.2.6 [46]. RAxML and MrBayes searches relied on the general time reversible (GTR) model of nucleotide substitution with the gamma model of rate heterogeneity. RAxML was run for the ML trees with 1000 bootstrap replicates. The Markov Chain Monte Carlo (MCMC) algorithm was calculated for 1,000,000 generations with a sampling tree every 1000 generations in Bayesian analysis. The first 25% of generations was discarded as burn-in. Stationary was reached when the average standard deviation of split frequencies was <0.01 and a consensus tree was constructed using the remaining trees.

## 3. Results

### 3.1. Chloroplast Genome Sequencing and Assembly

Using the Illumina sequencing platform, we sequenced cp genomes of nine *Camellia* species, including six species from the sect. *Oleifera*, two species from the sect. *Camellia*, and one from the sect. *Paracamellia* (Table 1). Illumina paired-end (2 × 150 bp) sequencing produced large data sets for individual samples. We randomly picked 25,000,000 paired-end reads to assemble the cp genomes. In total, 12 cp genome sequences were obtained through de novo genome assembly. Overall, 57,181~439,221 reads with an average insert size of 296~379 bp were assembled into the 12 cp genomes, reaching 112~884× average coverage (Table 2).

### 3.2. Characterization of Chloroplast Genomes of Selected Camellia Species

All 12 complete *Camellia* cp genomes exhibited a typical quadripartite structure of most angiosperms with conserved genome arrangement and structure. Among these cp genomes, genome size ranged from 156,545 bp (*C. sasanqua*) to 157,021 bp (*C. nanyongensis*). The sequenced genomes included a pair of inverted repeat regions (IRA and IRB) separated by a large single copy region (LSC) and a small single copy region (SSC) (Table 2 and Figure 1). The length varied from 86,257 bp (*C. sasanqua* and *C. chekiangoleosa* var. *Baihua*) to 86,659 bp (*C. oleifera* var. *40*, *53*, and *MY6*) in the LSC. It varied from 18,269 bp (*C.semiserrata*) to 18,415 bp (*C. chekiangoleosa* var. *Baihua*) in the SSC. The length varied from 25,943 bp (*C. meiocarpa*) to 26,060 bp (*C. nanyongensis*) in the IR (Table 2). These cp genome sequences were deposited into the GenBank under accession numbers OL689014~OL689025 (Appendix A).

Overall, the cp genomes of all 12 *Camellia* individuals encoded an identical set of 132 genes, including 87 protein-coding, 37 transfer RNA (tRNA), and eight ribosomal RNA (rRNA) genes (Table 2). Due to the uniform gene number, order, and names, the annotated cp genomes of these 12 *Camellia* taxa are represented in one circular map (Figure 1). In all cp genomes, eight protein-coding (*rps12*, *rps7*, *ndhB*, *ycf15*, *ycf2*, *rpl23*, *rpl2*, and *rps19*), seven tRNA (*trnN-GUU*, *trnR-ACG*, *trnA-UGC*, *trnI-GAU*, *trnV-GAC*, *trnL-CAA*, and *trnI-CAU*), and four rRNA (*rrn16*, *rrn23*, *rrn4.5*, and *rrn5*) genes were duplicated in the IR. The LSC harbored 61 protein-coding and 22 tRNA genes, while the SSC had 11 protein-coding and one tRNA gene. The *rsp19* is a uniquely structured gene with the 3’-end exon located in the LSC, while two copies of the 5’-end exon are located in the IRA and IRB, respectively; this arrangement generates an *rsp19* pseudogene in the IRA. The *ycf1* is located in the boundary region between the IRA and SSC, leading to an incomplete duplication of the gene within IRs and the generation of a pseudogene in the IRB.

All the 132 genes identified in the cp genomes could be divided into four categories according to their gene functions: photosynthesis-related genes, self-replication-related genes, other genes, and unknown function genes (Table 3). Furthermore, 18 intron-containing genes were found, including 16 genes (*trnK-UUU*, *rps16*, *trnG-UCC*, *atpF*, *rpoC1*, *trnL-UAA*, *trnV-UAC*, *rps12*, *petB*, *petD*, *rpl16*, *rpl2*, *ndhB*, *trnI-GAU*, *trnA-UGC*, and *ndhA*) containing one intron and two genes (*ycf3* and *clpP*) containing two introns (Table 3). Of the 18 intron-containing genes, 12 genes were located in the LSC, five genes were distributed in the IRs, while only the *ndhA* gene was in the SSC. Notably, the *trnG-UCC* gene of *C. chekiangoleosa* var. *Baihua* contained a longer Exon I (34 bp) and a shorter Exon II (43 bp) as compared to the other 11 *Camellia* individuals (23 and 48 bp, respectively); this observation could be regarded as a unique feature of *C. chekiangoleosa* var. *Baihua*. Of all the 20 introns identified, 12 introns exhibited length polymorphism with 1~26 bp across the 12 *Camellia* cp genomes (Table 4). The variance in the length of introns may be one of the reasons for the different sizes of these cp genomes.

### 3.3. Expansion and Contraction of the Border Regions

Although overall genomic structure, including gene number and order, is well-conserved, the expansion and contraction of the IRs are common in cp genomes, which is the main reason behind the different sizes of cp genomes [47]. The border regions and adjacent genes of the 12 genomes were compared to analyze the variation in expansion and contraction variation in the junction region (Figure 2). Results suggested that the junction positions of IR/LSC were conserved in these 12 genomes. For example, the *rsp19* gene, which was located at the boundary of the LSC/IRB in the cp genomes, showed the same length of 279 bp, including 233 bp in the LSC and 46 bp in the IRB, leading to incomplete duplication of the gene and the formation of a pseudo-rsp19 (46 bp) within IRA. There were slight differences in the junction positions of IR/SSC in these 12 genomes. The distances from *ndhF* to the junction of IRB/SSC ranged from 5~68 bp. Furthermore, the distances differed among the four *C. oleifera* varieties, i.e., 65 bp in *C. oleifera* var. *GR3*, rather than 68 bp in other three *C. oleifera* varieties. The distance was 68 bp in *C. meiocarpa* and *C. sasanqua*; 56 bp in *C. nanyongensis*, *C. vietnamensis*, *C. gauchowensis*, and *C. grijsii*; 62 bp in *C. chekianggoleosa* var. *Baihua*; and only 5 bp in *C. semiserrata* (Figure 2). *ycf1* gene was located on the junction of IRA/SSC with a length of 5616 bp in *C. meiocarpa*, *C. sasanqua*, and *C. chekianggoleosa* var. *Baihua*, rather than 5622 bp in the other nine cp genomes (Figure 2). The gene *ycf1* extended into the IRA with 963 bp in six cp genomes, 1069 bp in five cp genomes, and 1043 bp in *C. semiserrata*, leading to an incomplete duplication and formation of a pseudo-*ycf1* in the IRB with the corresponding length. Meanwhile, this pseudo-*ycf1* also extended into the SSC with 2 bp, 16 bp, or 30 bp, respectively (Figure 2). Notably, the length of *ycf1* in the IRA of *C. oleifera* var. *GR3* (1069 bp) was different from that in the other three *C. oleifera* varieties (963 bp). The variations at the IR/SSC boundary regions across the 12 *Camellia* cp genomes may lead to differences of the lengths of the IR and SSC and the whole-genome sequences.

### 3.4. Repeat Sequences and Microsatellite Analyses

In this study, we detected forward, palindromic, reverse, and complement repeats (long sequence repeats, LSRs) in all 12 *Camellia* cp genomes. Overall, 29~39 LSRs with lengths ≥30 bp were identified in each cp genome. Among these, 11~16 were forward repeats, 15~22 were palindromic repeats, and 0~2 were reverse repeats that were separately detected (Figure 3A and Appendix A). Only one complement repeat was screened in *C. chekiangoleosa* var. *Baihua*. The lengths of repeats in the 12 *Camellia* cp genomes ranged from 30 to 64 bp. The repeated lengths with 30~39 bp were the most common (48.58%), while those with 50~59 bp (10.43%) and over 60 bp (10.43%) were relatively rare (Figure 3B–D). Most of these LSRs were located in the exon region (57~71%), including some for protein-coding genes such as *ycf2*, *psbN*, *psaA*, and some tRNA genes. LSRs were also located in intergenic sequences (IGSs), while the least number of LSRs was present in introns (Figure 3E,F, and Appendix A). The LSRs were primarily shared over the cp genomes of eight species, where *C. nanyongensis* cp genome has the most unique LSRs (Figure 3G,H). It was found that *C. oleifera* var. *40*, *53*, and *MY6* shared a set of common LSRs, and *C. gauchowrnsis*, *C. sasanqua*, and *C. vietnamensis* shared another set of common LSRs (Figure 3H).

SSRs were also detected in the 12 cp genomes with repeat times of 3~20, and repeated lengths of 1~6 bp (Figure 4 and Appendix A). A total of 232~242 SSRs were detected in each genome. The majority of the SSRs were mononucleotide SSRs (especially for A/T) that varied from 142 in *C. meiocarpa* to 150 in *C. vietnamensis*. Trinucleotide SSRs (especially for the three-time repeat) was the second most predominant, ranging from 66 in *C. semiserrata* to 69 in *C. meiocarpa*, *C. nanyongensis*, and *C. sasanqua*. Furthermore, 10~11 tetra-, 4~5 di-, and 6~8 complex-nucleotide SSRs were identified in each cp genome. Additionally, two hexa-nucleotide SSRs were found in *C. oleifera*, *C. nanyongensis*, *C. chekiangoleosa*, and *C. semiserrata*, while the remaining five cp genomes did not contain any hexa-nucleotide SSRs (Figure 4, Table 5, and Appendix A). These SSRs were mainly located in the IGS, exons of some protein-coding genes, and tRNA genes, while only 13.56~14.66% of them were located in the introns. In the four structural regions, SSRs were distributed unevenly across the cp genomes, with the majority of SSRs were located in the LSC, followed by the SSC and IR (Table 5).

### 3.5. SNP and InDel Variations

Global alignment of the 12 *Camellia* cp genomes for the detection of SNPs and InDels was performed using *C. chekiangoleosa* var. *Baihua* as a reference. In total, 489 SNPs and 174 InDels were mined (Appendix A). The largest number of SNPs (40.90%) were distributed in the exon regions, followed by the IGS (33.33%) and intron regions (25.77%) (Appendix A). The base substitutions involving C and G were fewer than other types of substitutions, which is in agreement with previous studies [6,48]. The InDels were mainly distributed in the IGS (56.90%) and intron regions (33.33%), and only 9.77% were found in the exon regions (Appendix A). The short InDels (1~10 bp) accounted for 87.62% of total InDels (Appendix A). Similar to a previous study [6], single-nucleotide InDels were the most common and accounted for 37.62% of all InDels. The 5~6 bp InDels were more abundant than the 2~4 bp InDels and were the second most common type of all InDels characterized here (Appendix A). Notably, an InDel with a length of ~400 bp was detected at the intergenic regions of *atpH*-*atpI* (Appendix A). With *C. chekiangoleosa* var. *Baihua* as a reference, the *C. oleifera* var. *40*, *53*, *MY6* and *C. vietnamensis* had a 399 bp fragment insertion at the 15,054 bp loci, *C. oleifera* var. *GR3* had a 398 bp insertion, *C. grijsii*, *C. nanyongensis*, and *C. gauchowensis* had a 397 bp insertion. Notably, *C. semiserrata* had a 379 bp insertion at the 15,050 bp loci of the reference genome. *C. meiocarpa* and *C. sasanqua* did not present any insertion at this locus. Furthermore, we found two SNPs and three shorter InDels (1, 4, and 14 bp, respectively) in this insertion fragment of the eight *Camellia* taxa. This long InDels and its accompanying SNPs and shorter InDels may serve as candidate markers for phylogenetic reconstruction and species identification.

We performed a pairwise comparison of the cp genomes. The results showed that the number of SNPs and InDels ranged from 0 to 203 and 0 to 98, respectively, with the ratio of SNPs to InDels (S/I) ranging from 1.0 to 2.60 (Table 6). Three *C. oleifera* varieties shared an identical cp genome without any SNP and InDel, except for var. *GR3*. As compared to the intraspecific varieties and other species, var. *GR3* possessed similar numbers of SNPs and InDels. The results of the interspecies comparison showed that the cp genome sequences of *C. grijsii* and *C. gaochowensis* were highly similar and contained only two InDels and no SNPs, while the sequences of *C. nanyongensis* were different from the other species with more SNPs and InDels (Table 6). 

### 3.6. Sequence Divergence and Hotspots

To elucidate the level of genome divergence and identify the divergent hotspots, multiple alignments of 19 *Camellia* cp genomes, including 12 genomes sequenced in this work and seven genomes representative *Camellia* species from previous studies (Appendix A), were performed using Mauve [40]. With *C. chekiangoleosa* var. *Baihua* as a reference, we plotted the sequence identity (Figure 5). The alignment results revealed high sequence similarity across the 19 *Camellia* cp genomes, suggesting that the genome structure was highly conserved in terms of both gene identity and order (Figure 5). Differences were also detected across the *Camellia* cp genomes. Similar to the findings of a previous study [6], the IR were more conserved than single-copy regions, and coding regions were more conserved than noncoding regions. The most divergent regions were mainly located at positions between 5000~20,000 bp, 30,000~35,000 bp, 45,000~55,000 bp, and 115,000~135,000 bp (The cp genome sequence was coded in counterclockwise order from the interface between the LSC and IRA). Combined with the SNPs and InDels analysis results (Appendix A), the high variable regions were found that mainly located in *trnH-GUG- psbA*, *psbK-atpA*, *atpH-atpI*, *rpoB-psbM*, *trnE-UUC-psbD*, *rps4-ndhJ*, *ndhC-atpB*, *ndhF-rpl32*, and *ccsA-ndhD*. There was a marked divergence at the 15,065 bp position (*atpH-atpI* region) of the referent genome, which was in accordance with the existence of InDels with a length of ~400 bp at this locus (Figure 5 and Appendix A). These hotpots can be applied to DNA barcode encoding and phylogenetic analysis of *Camellia* genus.

Furthermore, the *Pi* values for each gene were calculated separately among the 19 *Camellia* cp genomes (Appendix A). The results showed that the distribution patterns of variable nucleotides in the IR/SC were fairly different (Figure 6A and Appendix A). The genes in the SSC showed the highest nucleotide diversity with a mean *Pi* value of 0.00116, followed by the LSC (mean *Pi* = 0.00072) and then the IR (0.00049). These findings illustrated that the IRs were more conservative than the single-copy regions. The top ten divergent hotspot genes were identified with *Pi* values. They were *psbK*, *psbT*, *psaJ*, *rpl33*, *rps8*, *rpl32*, *ycf1*, *rpl16*, *matK*, and *psbI* (Figure 6A and Appendix A). Among these divergent genes, eight were located in the LSC and two were located in the SSC. In these ten genes, four genes (*psbK*, *psbT*, *psaJ*, *psbI*) belonged to the ‘Subunits of photosystem’ group, four genes (*rpl16*, *rpl33*, *rpl32*, *rps8*) belonged to the ‘Proteins of ribosomal subunit’ group, and one gene (*matK*) encoded the maturase protein. These ten genes have a higher polymorphism level and can be applied to further phylogenetic analysis and species identification.

### 3.7. Estimation of the Selection Pressure of the Camellia Cp Genomes 

To evaluate the role of selection of the *Camellia* cp genomes, *Ka*, *Ks*, and *Ka*/*Ks* values of 87 protein-coding genes were calculated in 19 cp genomes using *C. chekiangoleosa* var. *Baihua* as a reference (Appendix A). The results indicated that the selection pressure of these genes was significantly different (Figure 6B). The *Ka*/*Ks* were not available for some genes because of the low level of divergence, and the *Ka* or *Ks* values were equal to zero. *Ka*/*Ks* values of most genes in each cp genome were less than 1, suggesting that the purifying selection may have acted on these protein-coding genes. Meanwhile, under the positive selection with *Ka*/*Ks* values >1, we identified four genes, *ropC2* in *C. gauchowensis*, *C. vietnamensis*, *C. grijsii*, and *C. cuspidata*; *petB* in *C.sinensis* and *C. nitidissima*; *ycf2* in *C. nanyongensis*; and *ycf1* in *C. vietnamensis*. *ropC2* and *ycf1* genes showed strong divergence and the *Ka*/*Ks* values were available for almost all cp genomes in this study (Figure 6B and Appendix A). The *ropC2*, *ycf1*, *ycf2*, and *petB* had sufficient potential to be used as barcode markers for phylogenetic analysis.

### 3.8. PCA and Phylogenetic Analysis of Camellia Species

We performed PCA to evaluate the relationships between the 12 cp genomes using SNPs data (Appendix A). Findings suggested that there was a substantial genetic diversity among the genomes (Figure 7). The species of sect. *Oleifera* were mixed within the other two sections (Figure 7A). Most species with the same geographical origin were grouped (Figure 7B). *C. oleifera* var. *40*, *53*, *MY6*, *C. meiocarpa*, *C. sasanqua* in sect. *Oleifera*; and *C. chekiangoleosa* var. *Baihua* in sect. *Camellia* were grouped with their geographical origin, i.e., Zhejiang, Jiangxi, and Fujian provinces of China, respectively. *C. oleifera* var. *GR3*, *C. nanyongensis* in sect. *Oleifera*, and *C. semiserrata* in sect. *Camellia* were clustered with the same geographical origin Guangxi Zhuang autonomous region of China. The last group involved *C. vietnamensis*, *C. gauchowensis* in sect. *Oleifera*, and *C. grijsii* in sect. *Paracamellia* with their common origin in Guangdong and Hunan provinces of China, respectively (Table 1 and Figure 7B).

Twelve cp genomes were sequenced in this study, combined with the complete cp genomes of 14 species from genus *Camellia* and six species from other genera of the Theaceae family published previously, and a total of 32 taxa were used to construct the phylogenetic trees using both the ML and Bayesian analysis methods (Figure 8, and the details of species shown in Appendix A). The phylogenetic relationships were evaluated within the genus *Camellia* and across the members of closely related genera based on the complete cp genome, coding region, and non-coding region data, respectively. The results showed that the six phylogenetic trees constructed were credible and most of their nodes were strongly supported by bootstrap or posterior probability values (≥70%) (Figure 8). The 32 taxa were classified into four major clades, of which all the *Camellia* species constituted a monophyly; *Adinandra* and *Anneslea* located at the basal position showed a close genetic relationship; *Stewartia* showed a large divergence from the other species and formed an independent clade, whereas *Pyrenaria*, *Tutcheria*, and *Apterosperma* clustered into another clade. This clade was a sister to the clade *Camellia* and showed a far genetic relationship to the clades formed by *Stewartia* and by *Adinandra* and *Anneslea*.

In the monophyly of *Camellia*, the phylogenetic relationships among 26 species and varieties were in general consistent across the six different phylogenetic trees, and the 26 taxa were divided into 3~6 distinct subclades (Figure 8). Subclade I was the largest one and involved 12~14 taxa, followed by subclade II with 7~9 taxa. The species and varieties of sect. *Oleifera* were predominant in subclade I and comprised a mix of three species (*C. chekiangoleosa* var. *Baihua*, *C. japonica*, and *C. azalea*) from sect. *Camellia*, *C. crapnelliana* from sect. *Furfuracea*, and *C. grijsii* from sect. *Paracamellia*. In subclade II, four species from sect. *Thea* were clustered into a branch and formed a sister group to the branches formed by *C. amplexicaulis*, *C. renshanxiangiae*, and *C. nanyongensis*. In the monophyly of *Camellia*, the six species from sect. *Camellia* were divided into several subclades, while most other species from the same sections were grouped (Figure 8). *C. semiserrata*, *C. nanyongensis*, and *C. semiserrata* had a rather distant phylogenetic relationship to the other taxa sequenced in this study, which is consistent with the results of PCA.

## 4. Discussion

### 4.1. Materials

As the largest and most important genus in the family Theaceae, the *Camellia* species have a great deal of economic importance, mainly due to their use in making beverages, ornaments, and edible oils. Regardless, the phylogenetic resolution of species in this genus remains a challenge. It has been suggested that cp genomes can suitably provide information for *Camellia* phylogenetic analyses [6,11,13,49]. In previous studies, detected taxa mainly focused on sect. *Thea*, while we performed phylogenetic analyses on plants from the sect. *Oleifera* by sequencing their cp genomes. We made this choice as these plants are widely grown for the production of high-quality edible seed oil in China. Meanwhile, *C. oleifera* has the most widely distribution in sect. *Oleifera* and formed several ecotypes. In this study, we selected four *C. oleifera* clones with geographical origins of central, southeast, and southwest regions of China, respectively. The other species samples were collected from one tree in the core of their geographical origin.

### 4.2. Genome Organization

Structural rearrangements and gene loss-and-gain events often occur in angiosperms. The *rpl23* gene translocates from the IR to the LSC regions in *Poaceae* plants [50]. *Phalaenopsis* and *Oncidium* lose most of their *ndh* genes [51,52]. It has often been reported that *ycf1*, *accD*, *rpl23*, *rps16*, and *ycf2* are absent in some plants [53,54,55]. The *Camellia* cp genomes show structural characteristics and genetic properties that are typical of the angiosperm plastomes. The genomic organization of *Camellia* in this study was well-conserved, where the gene order was identical to that in the *Diospyros*, *Artemisia*, and the other *Camellia* species [6,11,20,49,56]. The genome lengths of these taxa were ~157 kb on average. The genomes were organized into quadripartite regions with no structural variation among the taxa. Each genome encoded the same number of genes (87 protein-coding, 37 tRNA, and 8 rRNA genes), which included a pseudo-*ycf1* and a pseudo-*rps19* (Table 3 and Figure 2). We did not detect any gene loss-and-gain events in the 12 *Camellia* cp genomes.

IR expansion and contraction are commonly observed in plant cp genomes and are considered to be the primary reason for their length variations [47]. In *Poaceae*, for example, *ndhH* and *ndhF*, which are situated near the opposite ends of SSC, have been reported to migrate repeatedly into and out of the adjacent IRs [57]. IR expansions in monocot cp genomes lead to the duplication of the *trnH*-*rps19* gene cluster near the IR-LSC junctions [58]. Meanwhile, the *rps19* gene cannot be found in the cp genome of *Gossypium raimondii* because of IR contraction [59]. Similar to previous studies, the IR/SC boundary regions of 12 *Camellia* cp genomes studied here showed only slight differences (Figure 2) without remarkable IR expansions or contractions [6,11]. 

### 4.3. Sequence Divergence

Repeated sequences in plastome play important roles in genome rearrangement and are useful for phylogenetic studies [20,60,61]. Variations in the length and copy numbers of repeated sequences in the cp genome are widely used in plant population genetics, polymorphism investigations, and evolutionary research. This is especially important in well-conserved cp genomes of *Camellia*. In this study, we detected both the LSRs (≥30 bp) and SSRs. In contrast to the results of Yang et al. [11], the number of repeats in the 12 *Camellia* cp genomes matched well with that in other angiosperm [20,56,62]. The number and distribution of repeats were rather conserved among the *Camellia* cp genomes. The number of SSRs was more than LSRs. The majority of LSRs were located within exons (Appendix A), while SSRs were mainly located in IGS, followed by exons (Table 5). Most SSRs were mononucleotide repeats, where the A/T content was far greater than the G/C content (Figure 4), which is consistent with previous reports [6,62,63]. These repetitive sequences are important resources for studying differences in cp genes.

Previous research had suggested that SNPs and InDels play another important role in inducing variation [64,65]. We also detected SNPs and InDels in the whole cp genomes of the 12 *Camellia* taxa. SNPs were the most abundant variations, where 489 SNPs were identified in the 12 *Camellia* taxa (Appendix A). Generally, the number of InDels decreases rapidly with an increase in their length [66,67]. In line with this, we observed that single-nucleotide InDels (1 bp) were the most abundant InDel type. However, 5~6 bp InDels were more common than 2~4 bp InDels (Appendix A). Such 5~6 bp InDels were likely caused by adjacent 5~6 bp motif duplication or loss, making them more abundant than the shorter 2~4 bp InDels [6,59]. Some InDel mutations lead to frame-shift or a change in the codon sequence length in the protein-coding genes of *Camellia* cp genomes. A 5 bp InDel mutation that caused a frame-shift was identified in *rps16* in *C. semiserrata*. Two hexa-nucleotide deletion mutations made the *rpoC2* protein shorter by four amino acids in *C. semiserrata*. One hexa-nucleotide insertion mutation made the *rpoC2* protein longer by two amino acids in *C. chekiangoleosa* var. *Baihua* as compared to *rpoC2* protein in other *Camellia* taxa (Appendix A).

S/I values of taxa pairs ranged from 1.0 to 2.60 (Table 6), which is significantly lower than that for other species pairs from the *Camellia* genus [6]. This is consistent with the fact that the S/I values increase with divergence times between genomes [68]. In the 12 cp genomes studied here, *C. meiocarpa* was inferred to be the closest to *C. sasanqua* (S/I = 1.0), while *C. nanyongensis* could have diverged earlier, and thus, had higher S/I values (Table 6). *C. oleifera* var. *40, 53*, and *MY6* shared an identical cp genome, which suggested that cp genome sequences of *C. oleifera* ecotypes originated from the central and southeast of China were highly conserved and were different from that of southwest ecotypes.

### 4.4. Genome Divergent Hotspot Regions

Molecular evolutionary rates are often associated with life history in flowering plants [69]. The *Camellia* species possesses a low rate of molecular evolution because of the rather long generation times [6]. It is critical to be able to identify rapidly evolving cp genomic regions through comparative genomic analysis. Regions with relatively high sequence divergence can be developed into plant DNA barcodes. The noncoding *trnH*-*psbA* region contains LSRs that are highly variable in size and the sequence has been used as a barcode in the *Artemisia* plant [20,70]. *trnH*-*psbA* region in *Camellia* cp genomes contains multiple SNPs and had sufficient potential to be used as barcode markers (Appendix A). We found that *atpH-atpI* regions were highly variable regions in many plants, such as *Musa*, wild rice, wax gourd, *Iris*, etc., and can be applied to DNA barcode encoding and phylogenetic analysis [49,71,72,73,74]. In this study, we also found *atpH-atpI* region was highly diversity in the *Camellia* cp genomes, with multiple InDels and SNPs, which was consistent with the results of Huang et al. [6]. This region presents sufficient potential to be used as a barcode marker (Appendix A). Usually, most protein-coding genes in cp genomes have low sequence divergence in a limited number of species only [20,75]. Our research showed several potential barcodes in protein-coding genes, for example, the *rpoC2* gene harbored six various SSRs and three hexa-nucleotide InDels. Meanwhile, *psbK*, *psbT*, *psaJ*, *rpl33*, *rps8*, *rpl32*, *ycf1*, *rpl16*, *matK*, and *psbI* were identified as divergent hotspots based on the *Pi* values (Figure 6A and Appendix A). We speculate that positive selection was operating in the *rpoC2* and *ycf1* with the *Ka*/*Ks* values ≥ 1 in multiple *Camellia* taxa. Huang et al. indicated *rpoC2* and *ycf1* to harbor multiple SSRs in the sets of *Camellia* taxa and asserted that these SSRs could undoubtedly be subjected to assays for detecting polymorphisms at the population level and to compare more distant phylogenetic relationships at the genus level or above [6]. Previously, *rpoC2* gene sequence had been used to analyze the phylogenetic position of the genus *Astragalus* [76]. The *ycf1* coding sequence has also been suggested to be a plastid barcode for land plants and has been frequently applied in plant phylogeny and DNA barcoding studies [76,77,78,79,80,81,82]. *ycf1* plays an important role in flowering plants and is essential for plant viability [83]. According to previous studies, the high sequence divergence of *ycf1* in a range of land plants is likely a result of environmental adaptation during evolution [84,85,86], and thus, this gene may be used for the evolution analysis of cp. Li et al. also found the potential mutational hotspots genes *rbcL*, *matK*, and *ycf1* could be suitable barcodes for plant classification in *Camellia* [49].

### 4.5. Inference of Phylogenetic Relationships

Understanding phylogenetic relationships in the *Camellia* genus poses a grand challenge because of the high number of species, frequent hybridization, and polyploidization. Plastid phylogenomics offers a novel and deep insight into phylogenetic relationships and the diversification history [15,62]. In this study, we performed PCA on 12 cp genomes based on the SNPs data. Results showed that most taxa were grouped based on geographical origin, instead of section and species (Figure 7B). This may have been the result of interspecies hybridization based on the absence of recombination, as well as the maternal transmission characteristics of the cp genome.

We also constructed the phylogenetic trees of 12 genomes sequenced here and 20 complete cp genomes acquired from GenBank (Appendix A). Data from complete cp genomes, coding and noncoding regions were used for the purpose, respectively. The results were consistent with the traditional classification system of Chinese flora [5] at the genus level, while the intra-genus phylogenetic relationships of *Camellia* partially agreed with the traditional classification system. According to Sealy’s taxonomic treatment, most of the sect. *Oleifera* taxa had a closer phylogenetic relationship to each other, while *C. nanyongensis* had a closer relationship with *C. semiserrata*. Six species of sect. *Camellia* were divided into several subclades and mixed with the sect. *Oleifera* and *Thea*, respectively. The phenotypic characteristics of *C. chekiangoleosa* var. *Baihua* are similar to those of *C. chekiangoleosa*, except for the white flowers. However, it was closely related to *C. oleifera* var. *40*, *53*, and *MY6* according to the cp genome data. Whether *C. chekiangoleosa* var. *Baihua* is a hybrid of *C. chekiangoleosa* and *C. oleifera* still needs further investigation (Figure 8). 

Taxonomic studies on the *Camellia* genus are very controversial, especially on the definition of species and sections. The number of *Camellia* species and sections differ according to the different classification systems [1,5]. With the emergence of more complete cp genome sequences, the controversial taxonomy of *Camellia* can be addressed using cp genome analysis. Although this method is also insufficient to fully resolve all taxonomic controversies, our results suggest that it has the potential to provide a solution to many taxonomical disputes. With the rapid development of DNA sequencing technologies and a dramatic fall in the sequencing costs, complete cp genome analysis should be noted as a powerful tool to carry out taxonomic and phylogenetic studies in *Camellia*. The barcodes mined from the hotspot regions of cp genomes also have a significant potential to improve our ability to identify species or distinguish between them, and thus, can contribute to the phylogenetic relationships study. 

In recent years, many studies have used chloroplast whole-genome sequence for phylogenetic analyses and taxonomic classification [6,11,16]. This had been proven to be particularly powerful in revealing phylogenetic relationships of plants with large nuclear genomes [21,22]. Finding more regions with a higher evolution rate in the chloroplast genome sequence could be helpful to distinguish closely related species or genus, which is of great significance to the study of DNA barcodes [87].

## 5. Conclusions

The genome size, structure, and gene number and order were shown to be well-conserved among 12 cp genomes in this study. Meanwhile, there were slight differences in SSC/IR boundaries and in *ycf1* gene with different expansion lengths in different species. The whole plastome encoded 132 genes, including 87 protein-coding, 37 tRNA, and 8 rRNA genes. The cp genomes in this study possessed abundant variations, including LSRs, SSRs, SNPs, and InDels. A total of nine intergenic sequence divergent hotspots and 14 protein-coding genes with high sequence polymorphism were identified and had sufficient potential to be used as barcode markers in phylogenetic and taxonomic studies. PCA results showed that cp genomes had closer relationships with the same geographical origin rather than the same section. Phylogenetic trees suggested that *C. nanyongensis* had a rather distant phylogenetic relationship with the other taxa of sect. *Oleifera*. The complete cp genomes in this study will facilitate further research on *Camellia* and enhance our understanding of Theaceae family plastome evolution. It also provides a reference for the phylogenetic analysis of other plants.

## Figures and Tables

**Figure 1 biomolecules-12-01474-f001:**
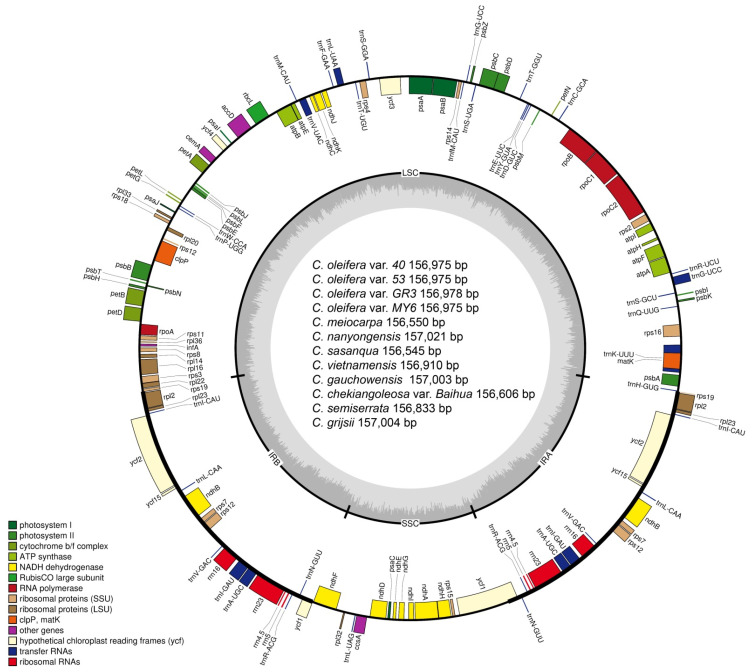
A gene map of the *Camellia* chloroplast genomes. The inner circle indicates the GC content, while the outer circle indicates the genome structure and gene arrangement. Genes shown outside the outer circle were transcribed clockwise and those shown inside were transcribed counterclockwise. Functional categories of genes are presented in the left margin.

**Figure 2 biomolecules-12-01474-f002:**
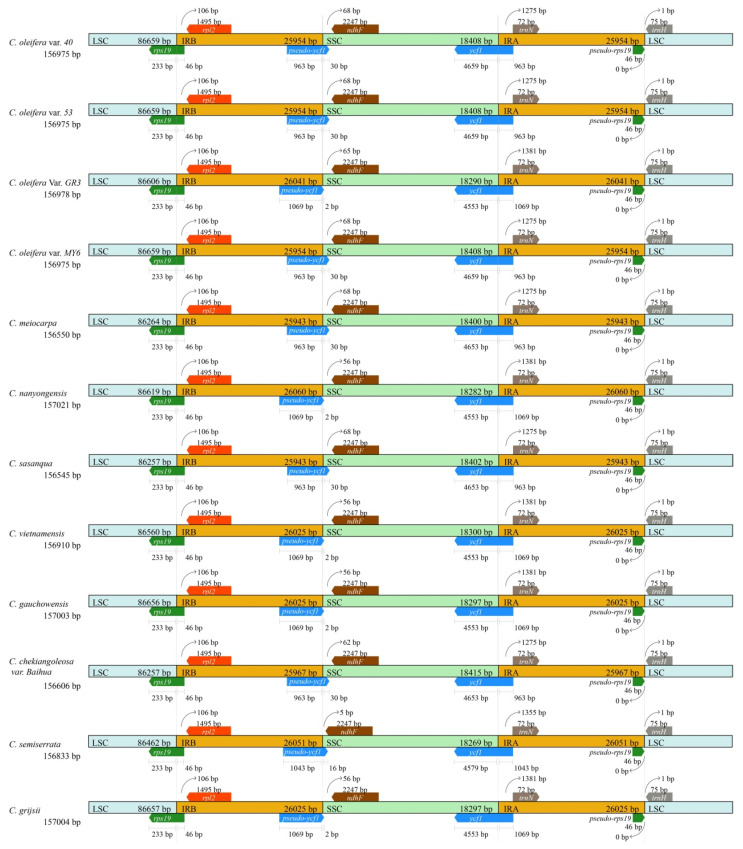
The comparison of the LSC, IR, and SSC border regions in the 12 *Camellia* chloroplast genomes.

**Figure 3 biomolecules-12-01474-f003:**
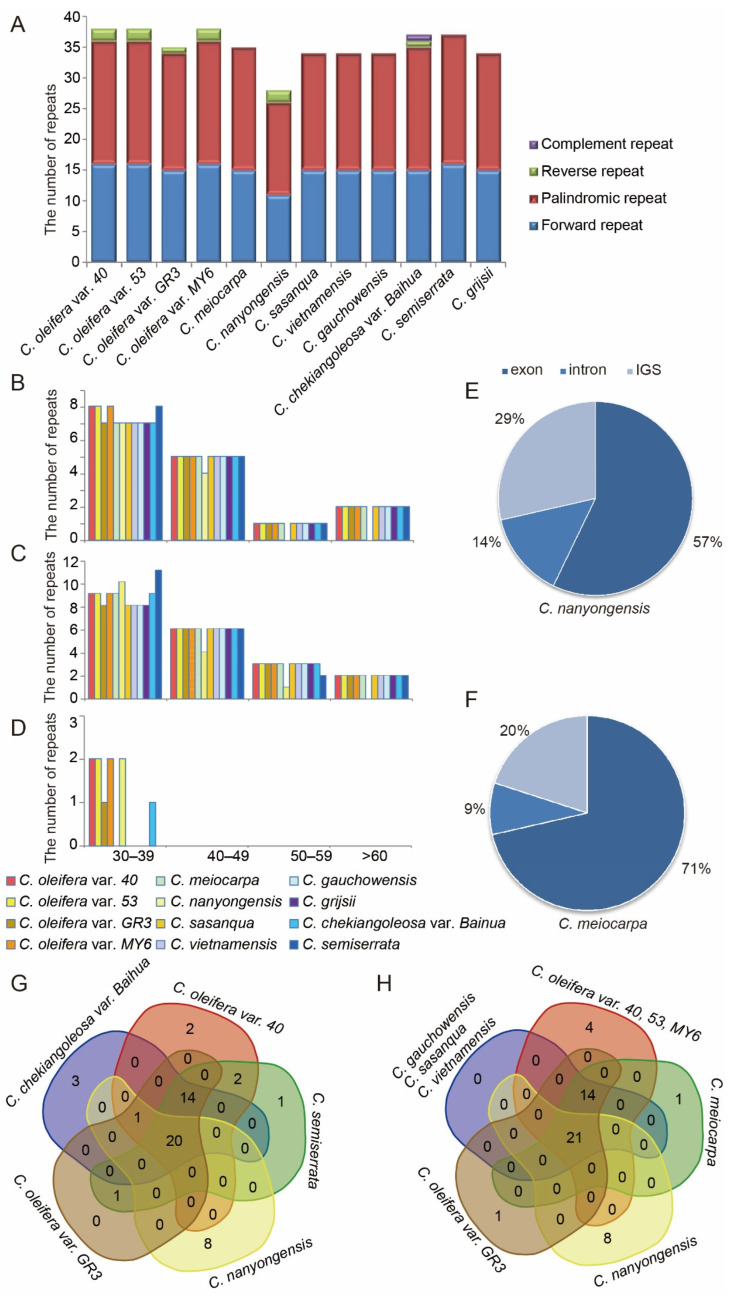
The type and distribution of LSRs in the 12 *Camellia* cp genomes. (**A**). Number of the four types of repeats; (**B**–**D**), Distribution of the forward, palindromic, and reverse repeats by length, respectively; (**E**,**F**), Location of repeats in *C. nanyongensis* and *C. meiocarpa*, which harbored the most and least repeats in exon regions in the 12 cp genomes, respectively; (**G**,**H**), Summary of shared LSRs among the 12 *Camellia* cp genomes.

**Figure 4 biomolecules-12-01474-f004:**
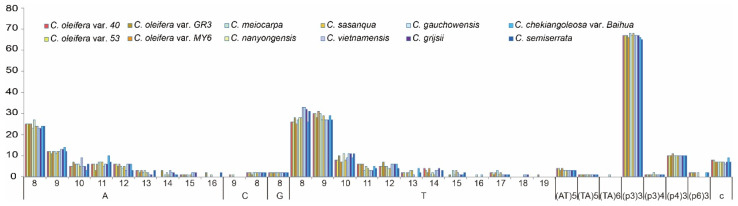
The distribution of SSR motifs in the 12 *Camellia* cp genomes. p3, p4, p6, and c indicate tri-, tetra-, hexa-, and complex nucleotides, respectively.

**Figure 5 biomolecules-12-01474-f005:**
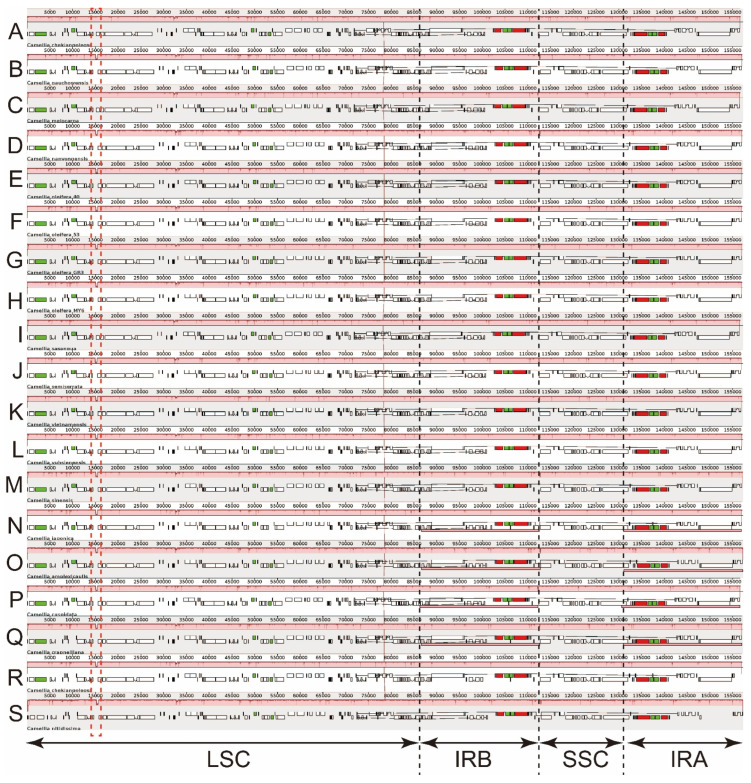
Mauve alignment of the 19 *Camellia* cp genomes. The *C. chekiangoleosa* var. *Baihua* genome is shown at the top as the reference genome. Within each alignment, the pink blocks represent the sequence similarity with the other genomes. The white, green, and red blocks represent protein-coding, tRNA, and rRNA genes, respectively. The regions in the red dotted box are the sequences with high divergence, which include the intergenic regions of *atpH*-*atpI*. A, *C. chekiangoleosa* var. *Baihua*; B, *C. gauchowensis*; C, *C. meiocarpa*; D, *C. nanyongensis*; E, *C. oleifera* var. *40*; F, *C. oleifera* var. *53*; G, *C. oleifera* var. *GR3*; H, *C. oleifera* var. *MY6*; I, *C. sasanqua*; J, *C. semiserrata*; K, *C. vietnamensis*; L, *C. grijsii*; M, *C. sinensis* (KC143082.1); N, *C. japonica* (MK353211.1); O, *C. amplexicaulis* (MT317095.1); P, *C. cuspidata* (NC022459.1); Q, *C. crapnelliana* (NC024541.1); R, *C. chekiangoleosa* (NC_037472.1); S, *C. nitidissima* (NC_039645.1).

**Figure 6 biomolecules-12-01474-f006:**
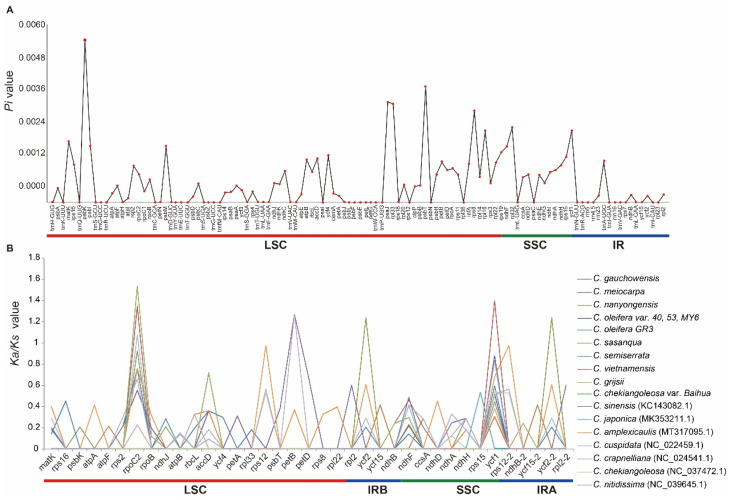
Nucleotide variability in each gene ((**A**); details in Appendix A) and selection pressure of protein-coding genes ((**B**); details in Appendix A) across the 19 *Camellia* cp genomes.

**Figure 7 biomolecules-12-01474-f007:**
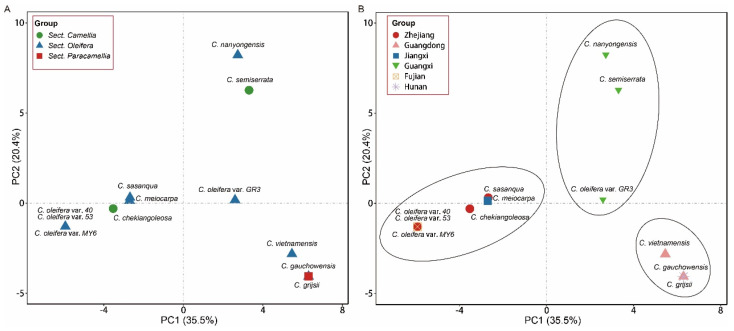
Principal component analysis plots of the 12 *Camellia* individuals. The meanings of each shape are indicated by the legend at the top.

**Figure 8 biomolecules-12-01474-f008:**
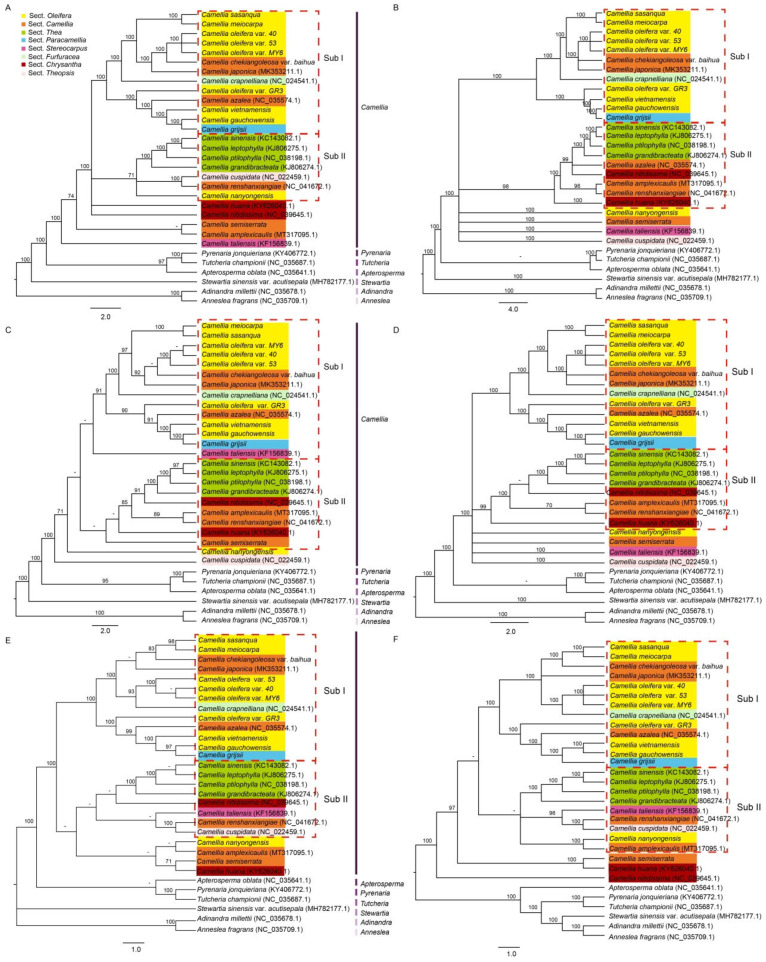
Phylogenetic trees of *Camellia* species and closely related genera in the family Theaceae. (**A**,**B**) Phylogenetic trees constructed using complete cp genome data with maximum likelihood and Bayesian analysis methods, respectively. (**C**,**D**) Phylogenetic trees constructed using the coding region data with maximum likelihood and Bayesian analysis methods, respectively. (**E**,**F**), Phylogenetic trees constructed using the noncoding region data with maximum likelihood and Bayesian analysis methods, respectively. Seven genera in Theaceae family are highlighted with vertical bars in different colors and eight sections in *Camellia* are marked with rectangles in different colors on the cladogram. The subclade I and II are marked with red dotted boxes. Numbers next to the branches correspond to maximum likelihood bootstrap support and Bayesian posterior probability values, respectively. The hyphen refers to the bootstrap or posterior probability values of <70%.

**Table 1 biomolecules-12-01474-t001:** Information of the sequenced oil–Camellia chloroplast genomes according to Sealy’s taxonomic treatment.

Taxon	Genus	Section	Collection Sites	Geographical Origin	Morphological characters
*Camellia oleifera* var. *40*	*Camellia*	*Oleifera*	DFFZP	Zhejiang province, China	Shrub or dungarunga; elliptical or ovate leaves; white flower, petals 5–7
*Camellia oleifera* var. *53*	*Camellia*	*Oleifera*	DFFZP	Zhejiang province, China	Shrub; elliptical or ovate leaves; white flower, petals 5–7
*Camellia oleifera* var. *GR3*	*Camellia*	*Oleifera*	DFFZP	Guangxi zhuang autonomous region, China	Shrub; elliptical or ovate leaves; white flower, petals 5–7
*Camellia oleifera* var. *MY6*	*Camellia*	*Oleifera*	DFFZP	Fujian province, China	Shrub; elliptical or ovate leaves; white flower, petals 5–7
*Camellia meiocarpa*	*Camellia*	*Oleifera*	DFFZP	Jiangxi province, China	Shrub; elliptical or ovate leaves; white flower, petals 5–7
*Camellia nanyongensis*	*Camellia*	*Oleifera*	ICSG	Guangxi zhuang autonomous region, China	Shrub; lanceolate leaves; white flower, petals 6
*Camellia sasanqua*	*Camellia*	*Oleifera*	RISF	Zhejiang province, China	Dungarunga; elliptical leaves; red flower, petals 6–7
*Camellia vietnamensis*	*Camellia*	*Oleifera*	RISF	Guangdong province, China	Shrub or dungarunga; elliptical or ovate leaves; white flower, petals 5–7
*Camellia gauchowensis*	*Camellia*	*Oleifera*	RISF	Guangdong province, China	Shrub or dungarunga; elliptical leaves; white flower, petals 7–8
*Camellia chekiangoleosa* var. *Baihua*	*Camellia*	*Camellia*	DFFZP	Zhejiang province, China	Dungarunga; elliptical leaves; white flower, petals 7
*Camellia semiserrata*	*Camellia*	*Camellia*	RISF	Guangxi zhuang autonomous region, China	Dungarunga; elliptical leaves; red flower, petals 6–7
*Camellia grijsii*	*Camellia*	*Paracamellia*	RISF	Hunan province, China	Shrub or dungarunga; oblong leaves; white flower, petals 5–6

RISF, Research Institute of Subtropical Forestry, Chinese Academy of Forestry (Hangzhou, China); ICSG, International *Camellia* Species Garden (Jinhua, China); DFFZP, Dongfanghong Forest Farm of Zhejiang Province (Jinhua, China).

**Table 2 biomolecules-12-01474-t002:** A summary of the statistics for the chloroplast genomic assemblies of 12 *Camellia* species and varieties.

Genome Features	*C. oleifera* var. *40*	*C. oleifera* var. *53*	*C. oleifera* var. *GR3*	*C. oleifera* var. *MY6*	*C. meiocarpa*	*C. nanyongensis*	*C. sasanqua*	*C. vietnamensis*	*C. gauchowensis*	*C. chekiangoleosa* var. *Baihua*	*C. semiserrata*	*C. grijsii*
Genome size (bp)	156,975	156,975	156,978	156,975	156,550	157,021	156,545	156,910	157,003	156,606	156,833	157,004
LSC size (bp)	86,659	86,659	86,606	86,659	86,264	86,619	86,257	86,560	86,656	86,257	86,462	86,657
SSC size (bp)	18,408	18,408	18,290	18,408	18,400	18,282	18,402	18,300	18,297	18,415	18,269	18,297
IR size (bp)	25,954	25,954	26,041	25,954	25,943	26,060	25,943	26,025	26,025	25,967	26,051	26,025
Number of genes	134	134	134	134	134	134	134	134	134	134	134	134
Protein coding genes (unique)	87 (80)	87 (80)	87 (80)	87 (80)	87 (80)	87 (80)	87 (80)	87 (80)	87 (80)	87 (80)	87 (80)	87 (80)
tRNA genes (unique)	37 (29)	37 (29)	37 (29)	37 (29)	37 (29)	37 (29)	37 (29)	37 (29)	37 (29)	37 (29)	37 (29)	37 (29)
rRNA genes (unique)	8 (4)	8 (4)	8 (4)	8 (4)	8 (4)	8 (4)	8 (4)	8 (4)	8 (4)	8 (4)	8 (4)	8 (4)
Duplicated genes in IR	34	34	34	34	34	34	34	34	34	34	34	34
GC content (%)	37.29	37.29	37.29	37.29	37.32	37.30	37.32	37.30	37.29	37.32	37.33	37.29
GC content in LSC (%)	35.29	35.29	35.30	35.29	35.33	35.32	35.33	35.31	35.29	35.34	35.34	35.29
GC content in SSC (%)	30.53	30.53	30.52	30.53	30.58	30.61	30.57	30.52	30.55	30.53	30.59	30.55
GC content in IR (%)	43.03	43.03	42.99	43.03	43.03	42.94	43.03	42.99	42.98	43.01	42.98	42.98
Total reads	25,000,000	25,000,000	25,000,000	25,000,000	25,000,000	25,000,000	25,000,000	25,000,000	25,000,000	25,000,000	25,000,000	25,000,000
Assembled reads	371,694	356,834	133,442	202,707	57,181	206,798	197,214	152,500	138,312	439,221	105,258	95,232
Average coverage	860	738	334	495	112	415	395	306	275	884	205	208
Average insert size (bp)	373	336	379	377	313	319	315	321	310	333	296	356

**Table 3 biomolecules-12-01474-t003:** List of annotated genes in the sequenced *Camellia* chloroplast genomes.

Category	Group of Genes	Name of Genes
Photosynthesis	Subunits of photosystem I	*psaA, psaB, psaC, psaI, psaJ*
	Subunits of photosystem II	*psbA, psbB, psbC, psbD, psbE, psbF, psbH, psbI, psbJ, psbK, psbL, psbM, psbN, psbT, psbZ*
	Subunits of NADH dehydrogenase	*ndhA *, ndhB *(2), ndhC, ndhD, ndhE, ndhF, ndhG, ndhH, ndhI, ndhJ, ndhK*
	Subunits of cytochrome b/f complex	*petA, petB *, petD *, petG, petL, petN*
	Subunits of ATP synthase	*atpA, atpB, atpE, atpF *, atpH, atpI*
	Large subunit of rubisco	*rbcL*
Self-replication	Proteins of large ribosomal subunit	*rpl14, rpl16 *, rpl2 *(2), rpl20, rpl22, rpl23(2), rpl32, rpl33, rpl36*
	Proteins of small ribosomal subunit	*#rps19, rps11, rps12 **(2), rps14, rps15, rps16 *, rps18, rps19, rps2, rps3, rps4, rps7(2), rps8*
	Subunits of RNA polymerase	*rpoA, rpoB, rpoC1 *, rpoC2*
	Ribosomal RNAs	*rrn16(2), rrn23(2), rrn4.5(2), rrn5(2)*
	Transfer RNAs	*trnA-UGC *(2), trnC-GCA, trnD-GUC, trnE-UUC, trnF-GAA, trnG-UCC, trnG-UCC *, trnH-GUG, trnI-CAU(2), trnI-GAU *(2), trnK-UUU *, trnL-CAA(2), trnL-UAA *, trnL-UAG, trnM-CAU, trnN-GUU(2), trnP-UGG, trnQ-UUG, trnR-ACG(2), trnR-UCU, trnS-GCU, trnS-GGA, trnS-UGA, trnT-GGU, trnT-UGU, trnV-GAC(2), trnV-UAC *, trnW-CCA, trnY-GUA, trnfM-CAU*
Other genes	Maturase	*matK*
	Protease	*clpP ***
	Envelope membrane protein	*cemA*
	Acetyl-CoA carboxylase	*accD*
	c-type cytochrome synthesis gene	*ccsA*
	Translation initiation factor	*infA*
Unknown function	Conserved hypothetical chloroplast ORF	*# ycf1, ycf1, ycf15(2), ycf2(2), ycf3 **, ycf4*

Notes: Gene *: Genes with one intron; Gene **: Genes with two introns; # Gene: Pseudo-gene; Gene (2): Two gene copies in the IRs.

**Table 4 biomolecules-12-01474-t004:** Information of intron-containing genes in the chloroplast genome of the *Camellia* species.

Gene	Location	Exon I (bp)	Intron I (bp)	Exon II (bp)	Intron II (bp)	Exon III (bp)
*trnK-UUU*	LSC	37	2488–2502	35		
*rps16*	LSC	39	851–875	225		
*trnG-UCC*	LSC	34 or 23	690 or 695	43 or 48		
*atpF*	LSC	159	704	408		
*rpoC1*	LSC	435	732	1626		
*ycf3*	LSC	126	737 or 747	228	722–725	153
*trnL-UAA*	LSC	37	523 or 519	50		
*trnV-UAC*	LSC	39	586	37		
*rps12*	IRa	346	538	26		
*clpP*	LSC	69	539–542	291	797–805	285
*petB*	LSC	6	756–762	657		
*petD*	LSC	9	696	525		
*rpl16*	LSC	9	1014–1025	402		
*rpl2*	IRb	393	667	435		
*ndhB*	IRb	777	679	756		
*trnI-GAU*	IRb	42	947 or 948	35		
*trnA-UGC*	IRb	38	812	35		
*ndhA*	SSC	552	1084–1110	540		

**Table 5 biomolecules-12-01474-t005:** Summary of SSRs in the 12 representative *Camellia* cp genomes.

Species	SSR Loci (N)	Region			Location			Styles					
		LSC	SSC	IR	Intron	Exon	IGS	P1 ^a^ Loci (N)	P2 ^b^ Loci (N)	P3 ^c^ Loci (N)	P4 ^d^ Loci (N)	P6 ^e^ Loci (N)	Pc ^f^ Loci (N)
*C. oleifera* var. *40*	237	139	50	48	33	88	116	145	5	68	10	2	8
*C. oleifera* var. *53*	237	139	50	48	33	88	116	144	5	68	10	2	8
*C. oleifera* var. *GR3*	238	139	51	48	33	89	116	147	4	68	10	2	7
*C. oleifera* var. *MY6*	237	139	50	48	33	88	116	145	5	67	11	2	7
*C. meiocarpa*	232	137	49	46	34	87	111	142	4	69	10	0	7
*C. nanyongensis*	242	142	52	48	33	89	120	149	5	69	10	2	7
*C. sasanqua*	233	137	50	46	34	88	111	143	4	69	10	0	7
*C. vietnamensis*	239	142	51	46	33	90	116	150	4	68	10	0	7
*C. gauchowensis*	237	140	51	46	33	90	114	149	4	68	10	0	6
*C. chekiangoleosa* var. *Baihua*	235	137	50	48	33	88	114	143	4	67	10	2	9
*C. semiserrata*	236	138	50	48	32	90	114	147	4	66	10	2	7
*C. grijsii*	237	140	51	46	33	90	114	148	4	68	10	0	7

**Table 6 biomolecules-12-01474-t006:** The numbers and ratios of SNPs and InDels in the 12 *Camellia* cp genomes.

	*C. oleifera* var. *40*	*C. oleifera* var. *53*	*C. oleifera* var. *GR3*	*C. oleifera* var. *MY6*	*C. meiocarpa*	*C. nanyongensis*	*C. sasanqua*	*C. vietnamensis*	*C. gauchowensis*	*C. grijsii*	*C. chekiangoleosa* var. *Baihua*	*C. semiserrata*
*C. oleifera* var. *40*	-	0	62	0	48	78	42	77	73	74	50	89
*C. oleifera* var. *53*	0	-	62	0	48	78	42	77	73	74	50	89
*C. oleifera* var. *GR3*	153(2.47)	153(2.47)	-	62	68	71	66	64	62	63	73	80
*C. oleifera* var. *MY6*	0	0	153(2.47)	-	48	78	42	77	73	74	50	89
*C. meiocarpa*	89(1.85)	89(1.85)	130(1.91)	89(1.85)	-	81	6	68	70	70	59	87
*C. nanyongensis*	203(2.60)	203(2.60)	175(2.46)	203(2.60)	178(2.20)	-	79	93	87	87	84	69
*C. sasanqua*	93(2.21)	93(2.21)	128(1.94)	93(2.21)	6(1.00)	176(2.23)	-	68	67	68	53	83
*C. vietnamensis*	170(2.21)	170(2.21)	121(1.89)	170(2.21)	143(2.10)	191(2.05)	147(2.16)	-	42	42	80	98
*C. gauchowensis*	171(2.34)	171(2.34)	122(1.97)	171(2.34)	147(2.10)	198(2.28)	151(2.25)	62(1.48)	-	2	77	94
*C. grijsii*	171(2.31)	171(2.31)	123(1.95)	171(2.31)	146(2.09)	198(2.28)	150(2.21)	62(1.48)	0	-	78	95
*C. chekiangoleosa* var. *Baihua*	75(1.5)	75(1.5)	140(1.92)	75(1.5)	72(1.22)	190(2.26)	76(1.43)	157(1.96)	158(2.05)	158(2.03)	-	91
*C. semiserrata*	194(2.18)	194(2.18)	173(2.16)	194(2.18)	173(1.99)	162(2.35)	177(2.13)	170(1.73)	173(1.84)	173(1.82)	183(2.01)	-

The upper triangle shows the number of InDels, and the lower triangle indicates the number of total SNPs. The ratios of SNPs to InDels (S/I) are given in brackets.

## Data Availability

The complete chloroplast genomes generated during the cuttent study were submitted to the NCBI database and are available with Genbank accession numbers OL689014~OL689025.

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
