# Peer review of "Comparative Genomic Analysis Uncovers the Chloroplast Genome Variation and Phylogenetic Relationships of Camellia Species"

_biomolecules, 2022, doi:10.3390/biom12101474_

Round 1

Reviewer 1 Report

This work describes comparative analyses of chloroplast (cp) genomes as a basis for phylogenetic analyses of Camellia species. Having in mind the controversy in taxonomic studies on the Camellia genus, complete cp genome analysis can present a powerful means underlying taxonomic and phylogenetic studies in Camellia.

Overall, the manuscript is very well written, with clear and concise representation of the results, complete and corresponding Discussion section, leading to conclusions with emphasis on the significance of the accomplished results. However, I have several minor remarks, and the comments regarding them are incorporated into pdf document, hopefully enabling easier approach for authors to response.

Author Response

Dear reviewer,

Thank for your patient and meticulous comments on our manuscript.

According to your comments, we removed the unnecessary data/graphs from the Table S2 excel file, revised the Table S3 for clarity and removed the table S8 in this revision.

We also corrected some grammatical issues to improve the writing of the manuscript, according your suggestions on revision. The details marked by red color can be found in this revision.

Reviewer 2 Report

The manuscript “Comparative genomic analysis uncovers the chloroplast genome variation and phylogenetic relationships of Camellia species” (biomolecules-1966585), showed the many results by chloroplast genomics in Camellia an important economical, ornamental, and popular plant in world.

The authors have done a good work, with many relevance phylogenetic analysis based an Illumina, which employing various references based on a scientific method and structure. The “big data” were analysed and proved our understanding of the sequencing and characterization of the chloroplast genome. The introduction, M&M, the results and discussion topic is good and minor points its necessary by adjusting in text. In addition, this manuscript demonstrated great and relevant results and discussion. However, the manuscript can be accept to Biomolecules journal after minor revision. Figure qualities were good, and acceptable for publication.

Please, use to template “Microsoft or Latex Template” of manuscript to figures, tables and references need adjust following “Author Instructions in Biomolecules”.

-Tables and figures, need correction following “Author Instructions” and displayed after first citation.

-Please, check all manuscript English language spelling.

Question

##01: How this methodology can be extended for other plants: potential challenges, advantages? Add in your discussion and complete in conclusion topic.

Minor points:

-Title: only the first capital letter of each word;

-Abstract, not bold;

-Alphabetic order keywords;

- Please. All standardization of nomenclature equipment/reagents/software when necessary. Example: Fabricant, City, State, Country (three-letter). Check all manuscript.

Best Regards

Author Response

Dear reviewers,

Thank you for the comment. In this revision, we have corrected the location of Tables and Figures following “Author Instructions”. Every tables and figures were displayed after the first citation. 

We have checked and corrected all spelling and grammatical issues. We also added the discussion about the potential challenges and advantages when this methodology was extended for other plants. Please find the details marked by read color in the revision.

In addition, we have revised all the minor points you mentioned and marked by read color in the revision.